# *Zika Virus*-Derived E-DIII Protein Displayed on Immunologically Optimized VLPs Induces Neutralizing Antibodies without Causing Enhancement of *Dengue Virus* Infection

**DOI:** 10.3390/vaccines7030072

**Published:** 2019-07-23

**Authors:** Gustavo Cabral-Miranda, Stephanie M. Lim, Mona O. Mohsen, Ilya V. Pobelov, Elisa S. Roesti, Matthew D. Heath, Murray A. Skinner, Matthias F. Kramer, Byron E. E. Martina, Martin F. Bachmann

**Affiliations:** 1The Jenner Institute, Nuffield Department of Medicine, Centre for Cellular and Molecular Physiology (CCMP), University of Oxford, Oxford OX1 2JD, UK; 2Immunology, RIA, Inselspital, University of Bern, 3010 Bern, Switzerland; 3Artemis Bio-Support, Molengraaffsingel, 2629 Delft, The Netherlands; 4Department of Chemistry and Biochemistry, University of Bern, 3010 Bern, Switzerland; 5Bencard Adjuvant Systems, Worthing BN14 8SA, UK; 6Department of Viroscience, Erasmus Medical Center, 3015 GD Rotterdam, The Netherlands

**Keywords:** vaccine, *Zika virus*, envelop (E) protein domain III (EDIII), virus like particles (VLPs), dioleoyl phosphatidylserine (DOPS)

## Abstract

*Zika virus* (ZIKV) is a *flavivirus* similar to *Dengue virus* (DENV) in terms of transmission and clinical manifestations, and usually both viruses are found to co-circulate. ZIKV is usually transmitted by mosquitoes bites, but may also be transmitted by blood transfusion, via the maternal–foetal route, and sexually. After 2015, when the most extensive outbreak of ZIKV had occurred in Brazil and subsequently spread throughout the rest of South America, it became evident that ZIKV infection during the first trimester of pregnancy was associated with microcephaly and other neurological complications in newborns. As a result, the development of a vaccine against ZIKV became an urgent goal. A major issue with DENV vaccines, and therefore likely also with ZIKV vaccines, is the induction of antibodies that fail to neutralize the virus properly and cause antibody-dependent enhancement (ADE) of the infection instead. It has previously been shown that antibodies against the third domain of the envelope protein (EDIII) induces optimally neutralizing antibodies with no evidence for ADE for other viral strains. Therefore, we generated a ZIKV vaccine based on the EDIII domain displayed on the immunologically optimized *Cucumber mosaic virus* (CuMVtt) derived virus-like particles (VLPs) formulated in dioleoyl phosphatidylserine (DOPS) as adjuvant. The vaccine induced high levels of specific IgG after a single injection. The antibodies were able to neutralise ZIKV without enhancing infection by DENV in vitro. Thus, the here described vaccine based on EDIII displayed on VLPs was able to stimulate production of antibodies specifically neutralizing ZIKV without potentially enhancing disease caused by DENV.

## 1. Introduction

*Zika virus* (ZIKV) is a mosquito-borne *flavivirus* transmitted to humans by infected *Aedes* mosquitoes [1,2]. In recent years, it was found that ZIKV may also be transmitted among humans without participation of the *Aedes* mosquitoes, for example by blood transfusion, maternal–foetal transmission, and sexually [3,4,5,6]. ZIKV is not a new virus and was first identified in 1947 in the Zika Forest of Uganda, Africa [1,7], with the first human infection reported in the 1950s. Before the outbreak in Brazil in 2015, ZIKV was not well known worldwide. Only thereafter, and when ZIKV infection became associated with microcephaly and cases of Guillain-Barré syndrome [8,9,10,11], did ZIKV call attention of the public as well as health authorities worldwide, and the World Health Organization (WHO) declared ZIKV as a Public Health Emergency of International Concern in 2016 [12].

ZIKV shares considerable genetic and structural similarity with other flaviviruses, for example, *Dengue virus* (DENV) [13], which is also transmitted by *Aedes* mosquitoes. Consequently, it might be presumed that the best strategy would be to develop a vaccine against all flaviviruses that circulate within the same ecological niche. However, this endeavour is complicated by the fact that poorly neutralizing antibodies that cross-react between several *flavivirus* types can cause a phenomenon called antibody-dependent enhancement (ADE). Such cross-reactive antibodies that induce ADE are particularly well described for DENV. They are poorly neutralizing but can enhance viral uptake and infection by the Fc receptor (FcR+) cells and consequently contribute to virus replication, which may lead to enhanced infection in vitro [14,15,16,17,18]. Clinically, it is well established that previous infection with a different DENV serotype may predispose to a more serious disease such as haemorrhagic fever. Even so, the mechanism that causes such disease enhancement is not completely clear, cross-reactive antibodies causing ADE in vitro and in preclinical mouse models are likely candidates. In addition, it is thought that secondary infection activate memory T cell responses, which may cause a cytokine storm and a more severe form of the disease, in particular in the presence of enhanced viral replication caused by cross-reactive ADE antibodies [17,19,20,21].

Most preclinical and clinical programs aimed to develop vaccines against ZIKV have focused on attenuated or inactivated viruses as well as viral and DNA-vectors [22,23,24,25]. Use of recombinant proteins or specific epitopes for vaccine development has gained less attention. The best example of antigens causing cross-reactive antibodies are the non-structural (NS), pre-membrane (PrM) and part of envelope (E) proteins, in particular the domain I (E-DI) and II (E-DII). In contrast, the E-DIII domain may be the best target for ZIKV vaccine development, as antibodies recognising this domain are largely specific for each *flavivirus* and/or serotype [19,26]. However, like other subunit antigens, E-DIII has a low inherent immunogenicity. For that reason, optimal epitope display and the use of adjuvants is crucial, as these factors play an important role in the potentiation of immunological responses induced by vaccines [27,28]. Repetitive display on virus-like particles (VLPs) is a potent way to enhance the immunogenicity of epitopes [29,30]. Recently we described a novel vaccine platform consisting of VLPs derived from *Cucumber mosaic virus* (CuMV), which was modified to incorporate a universal Th cell epitope derived from tetanus toxin (CuMVtt) to enhance antibody responses in individuals previously immunized against tetanus [31]. Immunogenicity of antigens displayed on VLPs is further enhanced due to the packaged RNA, which serves as a ligand for TLR7/8, stimulating B cells directly and driving IgG responses towards higher levels and more protective subclasses [32] as well as more potent secondary plasma cells [33].

There are many different adjuvant classes available. A recently described adjuvant is derived from Phosphatidylserine (PS), which is a natural and degradable component found on the inner side of plasma membranes and are translocated to the outer side when apoptosis occurs. PS serves as a surface signalling molecule that can flag apoptotic cells for recognition by phagocytic cells [34]. PS derivatives, in particular Di-oleoyl-phosphatidyl-serine (DOPS), have been shown to enhance specific antibody production and skew the response towards protective IgG subclasses [35]. CuMVtt displaying antigens derived from *Plasmodium falciparum* formulated in DOPS was shown to induce potent immune responses and provide protection against Malaria in preclinical mouse models [36].

In this study, we developed a vaccine-candidate against ZIKV by coupling E-DIII to CuMVttVLP and formulating the product with DOPS adjuvant. This vaccine was able to induce antibodies efficiently and neutralise the virus without predisposing for ADE with DENV infections. ZIKV and DENV are flaviviruses that circulate mostly in the same environmental niche and avoidance of disease enhancing antibodies is of critical importance. Therefore, this new vaccine candidate warrants further development.

## 2. Experimental Section

### 2.1. Zika Virus (ZIKV) E-DIII Protein Production

The cloning and production of ZIKV envelope protein domain III was published recently [37]. Briefly the E-DIII gene of *Zika virus* Brazil-ZKV2015 strain was selected from GenBank, reference number KY785450.1, and sent to be synthesised by Geneart™. Afterward, the gene was cloned into a pET-21(+) (Novagen, Merck, Nottingham, UK) vector with a C-terminal His6-tag and a short C-terminal cysteine (GGC). The new construct (ZIKV E-DIII into pET-21(+)) was transformed into BL21 (DE3) competent *Escherichia coli.* bacteria and the protein was expressed by induction with 1 mM isopropyl-β-d-thiogalactopyranoside into bacterial cultures. Subsequently, the *E. coli.* bacteria were collected by centrifugation, lysed and the inclusion bodies were washed and solubilized in 8 M urea, 100 mM Tris-Cl, 100 mM DTT, pH 8.0 and subsequently purified on a Ni Sepharose resin. The ZIKV ED-III protein, after purification, was refolded by dialysis. First dialysis was with 2 M urea, 50 mM NaH_2_PO_4_, 0.5 M arginine, 0.5 mM oxidized glutathione, 5 mM reduced glutathione, 10% glycerol, pH 8.5; followed with dialysis against 50 mM NaH_2_PO_4_, 0.5 M arginine, 0.5 mM oxidized glutathione, 5 mM reduced glutathione, 10% glycerol, pH 8.5, and finally dialysis with 50 mM NaH_2_PO_4_, 10% glycerol, pH 8.5. Pure protein was obtained by removing residual contamination with size-exclusion chromatography (SEC) using a HiPrepTM16/60 SephacrylTMS166HR column (GE Healthcare, Glattbrugg Switzerland).

### 2.2. Vaccine Formulation: Coupling CuMVttVLP with E-DIII and Mix with Di-Oleoyl-Phosphatidyl-Serine (DOPS)

The CuMVttVLP production was previously described in detail [31]. Concisely, the CuMVttVLP was expressed in *E. coli.* BL21 (DE3) Star (Thermo Fisher Scientific, Loughborough, UK) and purification was done by using the soluble part of the cell lysates. The purified CuMVttVLP was covalently conjugated with E-DIII protein by previously adding 1 mg/mL of VLP in 20 mM sodium phosphate, 2 mM EDTA, and 30% (*w*/*v*) sucrose, at pH 7.2 (PES buffer) and incubated at room temperature for 30 min. At the same time, the chemical cross-linker SMPH (succinimidyl-6-(b-maleimidopropionamido) hexanoate) (Thermo Fisher Scientific) was added at a concentration of 10-fold molar excess. Afterward, the unreacted SMPH was removed by diafiltration. In parallel to linking CuMVttVLP with SMPH, the E-DIII protein was incubated with *N*-succinimidyl-S-acetylthioacetate (SATA) for 30 min at room temperature. The SATA excess that did not react with E-DIII was removed by diafiltration and then hydroxylamine was added and incubated for three hours at room temperature to induce reactive sulphydryl residues in the E-DIII protein. After that, the protein was once more diafiltrated and covalently linked to CuMVttVLP-SMPH by mixing with an equimolar amount and incubated for four hours at room temperature [38]. Atomic force microscopy (AFM) images were obtained with CuMVttVLP only and with E-DIII displayed on it. For that, the AFM imaging was carried out in an ambient environment at room temperature employing Nanosurf FlexAFM scan head (100 μm scan range) with C3000 controller using PPP-NCHAuD cantilevers (Nanosensors) in dynamic mode. Obtained images were processed using Gwyddion software.

The vaccine (CuMVttVLP-EDIII) formulation using DOPS adjuvant was prepared by using 50 µg of DOPS per dose per mouse. For that, the DOPS was dissolved in PBS and mixed with CuMVttVLP-EDIII and gently vortexed for a few seconds and immediately used for vaccination [36].

### 2.3. Ethics Statement and Animal Use

These experiments were performed using female inbred BALB/c mice age-matched 6 to 8 weeks old (mature immune system) and all procedures were done in agreement with the terms of the UK Home Office Animals Act Project License. The procedures were approved by the University of Oxford Animal Care and Ethical Review Committee (PPL P9804B4F1).

### 2.4. Schedule of Immunization

For assessment of vaccine immunogenicity, five female BALB/c inbred mice per group were vaccinated intramuscularly (i.m.) with 20 µg (50 μL) per mouse of CuMVttVLP-EDIII vaccine, CuMVttVLP-EDIII formulated with DOPS adjuvant, only E-DIII protein diluted in phosphate-buffered saline (PBS) or only PBS as a negative control. The vaccinations were performed three times, with 21 days between each injection. Blood samples were collected by tail vein puncture before each vaccination, on Days 0, 21, 42 and 63, to check the antibody response. At the end of each experiment all mice were humanely sacrificed by an approved Schedule 1 method (cervical dislocation). Two independent experiments with five female mice per arm were performed, totalizing 10 mice per group.

### 2.5. Assessment of IgG Total and IgG Subclass Titers

The production of IgG antibodies was analysed by enzyme-linked immunosorbent assays (ELISAs). The ELISA plates (Thermo Scientific, Nottingham, UK) were coated with 100 µL of E-DIII protein at a concentration of 1 µg/mL previously diluted in 50 mM carbonate buffer (CBB) and incubated overnight at 4 °C. The following day the ELISA plates were washed three times with PBS-0.05% Tween and subsequently coated with 200 µL of PBS Casein 0.5% as a blocking agent. After two hours of blocking, the sera mice samples were added and a serial dilution was performed, first with a 1 in 100 dilution and then eleven 1:3 dilutions. As a secondary antibody, goat anti-mouse total IgG HRP conjugate (ThermoFisher, Paisley, UK) was added. The specific IgG subtypes used were goat anti-mouse IgG1, IgG2a and IgG2b HRP coupled (ThermoFisher, Paisley, UK). The antibody dilution was 1:2000 and incubated for two hours at room temperature. The reaction was developed by adding 100 µL per well of TMB substrate (Sigma-Aldrich/ Merck, Darmstadt, Germany) and incubated for 10 min in the dark, and then stopped with 0.5 M H_2_SO_4_ (sulfuric acid) before reading the plates using a microplate reader at 450 nm. The antibody titers are shown as dilutions leading to half-maximal OD (OD_50_) and the values observed in the negative control group were subtracted from the titres of the vaccinated groups.

### 2.6. Zika Virus Neutralization Assay

The test of neutralizing capacity of the antibody anti-ZIKV EDIII protein was performed using Vero cells (ATCC^®^ CCL-81™, Manassas, VA, USA). The cells were cultured in Dulbecco’s modified Eagle medium (DMEM) with 10% heat-inactivated foetal bovine serum (HI-FBS; Lonza Benelux BV, Breda, The Netherlands), supplemented with 0.75% sodium bicarbonate and 10 mM hepes buffer (Lonza) until a confluent monolayer was obtained. Serum from vaccinated mice was heat-inactivated at 56 °C for 30 min to inactivate complement, and was serially dilutted two-fold, starting at 1:10 to 1:80. Subsequent incubation with an equal volume of Zika-Padova 1/201, Vero passage 2 [39] resulted in a final serum dilution of 1:20 to 1:160, and a virus concentration of 400 TCID_50_/well. The serum-virus mixtures were incubated at 37 °C for 1 h and subsequently added to a 100% confluent monolayer of Vero cells in CELLSTAR^®^ 96-well plates (Greiner Bio-One, Alphen aan den Rijn, The Netherlands). Plates were incubated at 37 °C for 5 days. Samples were read and a 100% reduction in cytopathic effect (CPE) as compared to the serum-negative control was used for the determination of neutralization. The serum samples were tested in two independent experiments.

### 2.7. Antibody-Dependent Enhancement Test with Dengue Virus (DENV-2)

The ADE test was performed using THP-1 cells, obtained from ATCC^®^ and cultured with RPMI 1640 supplemented with 50 μM beta-mercaptoethanol, penicillin (100 U/mL, Lonza), streptomycin (100 μg/mL, Lonza), and 10% foetal bovine serum (FBS; Gibco/ ThermoFisher, Paisley, UK) (R10F) at 37 °C in 5% CO_2_ in a humidified incubator. The ADE assay was conducted with unstimulated as well as with stimulated/differentiated THP-1 cells. The mature macrophage-like state was induced by treating THP-1 cells (10^5^ cells/mL) with 50 ng/mL phorbol 12-myristate 13 acetate (PMA) for 72 h. Subsequently, the cells were maintained for 48 h in fresh R10F lacking PMA. After that, serial two-fold dilutions of the sera from vaccinated mice were made and dengue virus type 2/NGC strain (P7) was added at a multiplicity of infection (MOI) 1. The pan-*flavivirus* monoclonal antibody and pool of serum from mice infected with rabies virus were used as controls. The mixture was incubated for 2 h at 37 °C and subsequently incubated with unstimulated (10^5^) and stimulated cells. After an incubation period of 2 h, the mixture was removed, cells washed twice with R10F and further incubated in fresh medium for another 48 h. To measure ADE, RNA was isolated from the supernatant and a quantitative real-time polymerase chain reaction (qRT-PCR) was conducted as previously described [40]. Briefly, a one-step qRT-PCR was performed using primers and probes directed against the 3′UTR, derived from Drosten et al. [41]. TaqMan Fast Virus 1-step Master Mix (4×) was used with 15 pmol of primers and 10 pmol of probes and an additional 25 mM of MgCl2 was added. The cycling program consisted of 5 min at 50 °C, then 20 s at 95 °C followed by 40 cycles of 3 s at 95 °C and 30 s at 60 °C. The RNA copy numbers in each sample were calculated from a standard curve generated from run-off 3′-UTR transcripts.

### 2.8. Statistical Analysis

The statistical analysis was preformed using GraphPad Prism software version 5.0 for Max OS. When comparing two normally distributed groups a unpaired *t*-test was used; when more than two groups were analysed, a Kruskal–Wallis test with Dunn’s multiple comparison post-test was performed, whereas normally distributed data were analysed by a one-way analysis of variance (ANOVA) with Bonferroni’s multiple comparison post-test. The value of *p* < 0.05 was considered statistically significant (* *p* < 0.01, ** *p* < 0.001, *** *p*< 0.0001).

## 3. Results

### 3.1. Choosing the Target Epitope to Design a ZIKV Vaccine

The development of vaccines against flaviviruses represents a particular challenge, as antibodies that are poorly neutralizing but cross-reactive between different *flavivirus* species and/or serotypes may cause enhanced viral replication. Therefore, it is essential to use epitopes for immunization that maximize neutralization and minimize cross-reactivity. It has been shown for DENV that domain III of the envelope proteins (EDIII) is a target for strongly neutralizing and poorly cross-reactive antibodies. Consequently, EDIII-specific antibodies should not cause ADE. The same observation has recently been made for ZIKV, where again the EDIII-domain was recognized by highly neutralizing, but poorly cross-reactive antibodies not causing ADE [16,26]. Using sensitive biosensors, we have shown that antibodies derived from DENV-infected patients did not recognize the EDIII-domain of ZIKV, further supporting the specificity of the EDIII-specific antibodies [37]. Furthermore, we have previously shown that the EDIII-domain of *West Nile virus* (WNV), another *flavivirus*, induced highly neutralizing and protective antibodies in mice when displayed on VLPs [42].

We therefore expressed the EDIII of ZIKV with a small linker containing a free Cys for chemical coupling in *E. coli.* [37] and displayed it on the CuMVttVLP surface using a chemical crosslinker (SMPH) (Figure 1a). We analysed CuMVttVLP alone (Figure 1b), as well as CuMVttVLP coupled to E-DIII (Figure 1c) by Atomic force microscopy (AFM). To further enhance immunogenicity, the vaccine-candidate was formulated in DOPS, which has previously been shown to enhance immunogenicity of CuMVtt-based vaccines [36].

### 3.2. Humoral Immunity

To assess the humoral immune response induced by the vaccine candidate, mice were immunized 3 times on day 0, 21 and 42 with EDIII alone, CuMVtt-EDIII or CuMVtt-EDIII formulated in DOPS. Antibody production was assessed by ELISA using the sera collected three weeks after the prime vaccination (Day 21), three weeks after the first boost (Day 42) and three weeks after the second boost (Day 63). As shown in Figure 2, the best vaccine formulation was CuMVtt-EDIII + DOPS that induced the highest antibody response followed by CuMVtt-EDIII formulated in PBS, and EDIII alone. Both CuMVtt-EDIII formulations were able to induce strong IgG responses after a single immunization.

Analysis of IgG subclasses revealed that DOPS suppressed IgG1 responses and enhanced IgG2a and IgG2b responses, corroborating results obtained with a previous vaccine formulations against Malaria [36]. In contrast, EDIII alone only induced IgG1 but failed to generate IgG2a and IgG2b responses. CuMVtt-EDIII induced an intermediate response, and all three IgG subclasses were generated (Figure 3).

### 3.3. Neutralization Capacity of Antibodies

A critical test to assess the efficacy of a vaccine candidate is to assess whether it is able to induce neutralising antibodies that prevent viral infection. To this end, we have performed a CPE-based neutralization assay using a representative ZIKV strain inducing good CPE (Figure 4).

As shown in Figure 5, the group of mice vaccinated with VLP-EDIII+DOPS developed antibody responses able to neutralise the ZIKV after two vaccinations (Day 42). In contrast, mice vaccinated with VLP-EDIII without adjuvant generated humoral immunity able to neutralise the virus only after three vaccinations; at a similar level seen with DOPS formulated vaccine after two injections. Thus, DOPS enhances both the speed and the magnitude of the response. In addition, EDIII protein alone without VLP-conjugation and adjuvant was not able to induce measurable levels of neutralizing antibodies.

### 3.4. Assessment of ZIKV Antibody-Dependent Enhancement with DENV-2

Enhanced infection due to cross-reactive antibodies is well characterized for DENV. DENV-2 is considered to be the most virulent DENV serotype with a genetically diverse population and is most frequently associated with dengue epidemics worldwide. More importantly, most of the epidemics of dengue haemorrhagic fever have been associated with DENV-2 instead of the other serotypes. Therefore, the use of DENV-2 served as an excellent model for this study [43]. As ZIKV-induced antibodies may cross-react with DENV, a key feature of a ZIKV vaccine candidate is that it must not induce antibodies that enhance DENV infection. As described in the schematic figure (Figure 6a,b) representing a specific response against a virus, when it has been completely opsonised by neutralizing antibody (Figure 6a), the phagocytic cells containing Fcy receptor will bind the Fc part of the antibody and internalise the IgG complexed virus. After internalisation, phagolysosomes will be formed and will destroy the virus. However, as shown in Figure 6b, when ZIKV is recognised by non-neutralizing antibodies, the antibody binding will not be able to neutralise the *flavivirus*, but instead it will help the viral uptake without destruction via phagolysosomes, allowing therefore the cellular infection. Consequently, the virus can replicate to higher levels, likely causing increased disease. Therefore, to assess whether the vaccine candidate developed here would enhance DENV-infection in vitro, we performed an ADE-test for the vaccine-induced antibodies using DENV-2. To this end we used the THP-1 monocytic cell line that are derived from acute monocyte leukemic patients [44] and has been extensively applied to assess the mechanism of ADE with DENV-2 infection. Part of the THP-1 cells were stimulated to obtain a mature macrophage-like state. Serial dilution of the sera, obtained from mice three weeks after the third vaccination (Day 63), were made and both unstimulated and stimulated cells were infected at a multiplicity of infection of DENV-2/NGC strain (P7). As seen in Figure 6c,d, the qRT-PCR results demonstrate that ZIKV specific serum antibodies induced by vaccination did not promote the synthesis of more DENV RNA as measured in the supernatants. Under none of the conditions tested (stimulated or unstimulated THP-1 cells), an enhancement of infection and viral replication was measured. When comparing the positive control with all other groups, Negative control, CuMVtt-EDIII or CuMVtt-EDIII + DOPS, it showed to be highly significant (*p* value < 0.0001). Therefore, in addition to inducing virus-neutralizing antibodies, this vaccine-candidate also passed the test of not inducing ADE antibodies.

## 4. Discussion

The widespread outbreak of ZIKV in the Americas and its causal association with severe congenital disease stimulated research on the development of save and effective vaccines. Several platforms were evaluated especially for their ability to stimulate ZIKV-specific, non-enhancing antibodies. Several studies have shown that immune responses to ZIKV were similar to responses induced against other flaviviruses such as DENV or WNV. Specifically, it has been shown that neutralising antibodies are directed primarily against the E protein and are associated with protection against infection [45,46]. However, due to the extensive cross-reactivity between flaviviruses, cross-reactive antibodies may result in enhancement of infection or disease with ZIKV or DENV [18,47]. Therefore, the issue of enhancement has to be considered when developing vaccines against *flavivirus*, such as ZIKV and DENV [14,15,16,17,18,19,20,21]. In this study we have shown that a VLPs vaccine candidate carrying ZIKV E-DIII induced neutralizing antibodies, but no enhancement against DENV-2 was observed.

The number of vaccine candidates against ZIKV have increased recently and some of them have advanced to clinical trials. Three types of those vaccines that achieved phase 1 clinical trials are based on DNA, modified RNA and inactivated virus [22]. The E protein of flaviviruses (including ZIKV) can be divided into three domains, with most neutralising antibodies targeting determinants in DIII or the fusogenic loop of DII [48]. Therefore, the use of specific domains or epitopes may hold the key for success. For ZIKV, particular concern is warranted as infections of pregnant woman can cause induction of microcephaly in babies [8,9,10,11]. Current vaccine candidates, which are based on whole inactivated virus, or the complete E-protein may induce non-neutralizing and cross-reactive antibodies that could enhance disease of either ZIKV or DENV. Vaccines based on E-DIII are likely to circumvent this problem as E-DIII induces highly specific neutralizing antibodies and low levels of cross-reacting and enhancing antibodies. Indeed, a recent publication has demonstrated that Zika EDIII-domain fused to hepatitis B core Antigen (HBcAg) induces potent neutralizing antibodies in mice without enhancing DENV virus infection in vitro [49].

Immunogenicity of the candidate ZIKV vaccine was tested in Balb/C mice. We wanted to understand the subclass distribution since that would indirectly indicate the type of Th cell (Th1 versus Th2) response induced by the vaccine. IgG1 responses are usually associated with Th2 responses, whereas high levels of Ig2a and IgG2b are believed to reflect Th1 responses [50]. The response elicited by the CuMVttVLPs-EDIII candidate vaccine showed high levels of multiple IgG1 subclass, indicating a Th2 response, whereas the presence of the adjuvant (DOPS) appeared to strongly regulate IgG2b production, reflecting adjuvant-dependent Th1 responses. It is interesting to note that IgG1 represents the predominant subclass generated by most adjuvants independent of the mouse strain. Our data support a modulatory effect of the adjuvant on the Th1/Th2 balance.

The differential potential of different IgG subclasses to bind and activate the complement system is well described [51]. Several studies have shown that complement could neutralize several flaviviruses in an antibody-dependent and independent way [52,53]. Therefore, it is conceivable that certain IgG subclasses could trigger complement-dependent neutralization and therefore result in higher neutralizing titers. We have shown that complement alone can inactivate ZIKV and DENV (unpublished) and, therefore, we did not study complement-dependent neutralization in our study. Clearly, more efforts should be deployed to develop assays, which allow discrimination of IgG subclass dependent and independent complement activation and neutralization.

Our candidate vaccine did not induce enhancing antibodies against DENV-2. However, our data solely indicate that the ZIKV-induced antibodies did not enhance the NGC strain of DENV-2. It has been suggested that ADE may be serotype- and genotype-dependent [54,55,56], so we cannot exclude that infection of cells with other serotypes or genotypes could be enhanced by the ZIKV antibodies. Therefore, studies investigating enhancement activity of *flavivirus* cross-reactive antibodies must include a panel of DENV-serotypes and genotypes. Taken together, our data show that CuMVttVLP-EDIII induces strong antibody response and that the use of adjuvant can skew the immune response in a Th1 direction.

## Figures and Tables

**Figure 1 vaccines-07-00072-f001:**
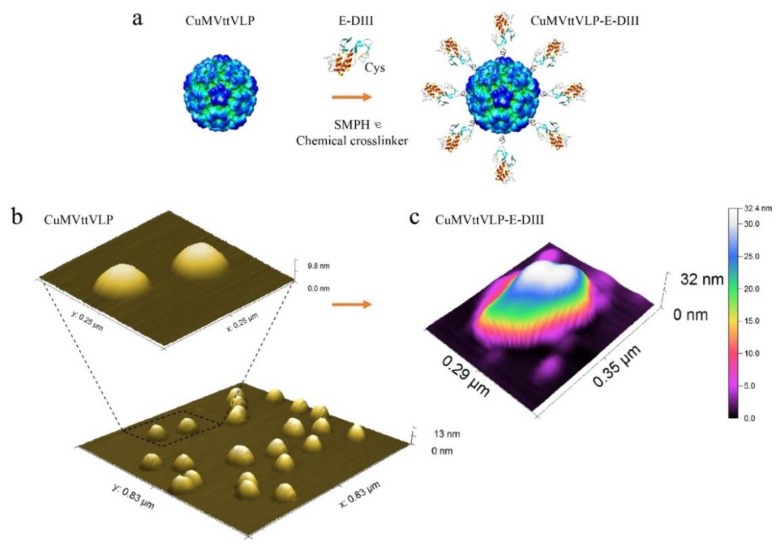
Summarizes the vaccine design and atomic force microscopy (AFM) images. (**a**) The conjugation of Zika virus E-DIII protein was done by modifying the CuMVttVLP with a chemical crosslinker (SMPH) and binding it to a modified protein with sulphydryl groups (-SH), as described in the Methods section “Vaccine formulation: coupling CuMVttVLP with E-DIII and mix with dioleoyl phosphatidylserine (DOPS)”. The AFM imaging was carried out by employing Nanosurf FlexAFM scan head (100 μm scan range) with C3000 controller using PPP-NCHAuD cantilevers (Nanosensors) in dynamic mode and processed using Gwyddion software. (**b**) Yellow samples show the CuMVVLP alone, without any protein attached, and (**c**) the following image shows a single particle with E-DIII proteins attached to it. The colour scale was adjusted to visually enhance smaller objects.

**Figure 2 vaccines-07-00072-f002:**
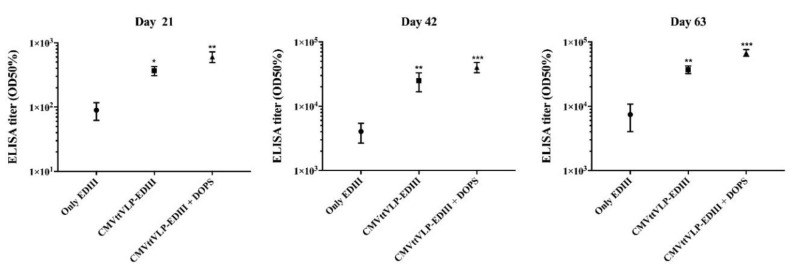
CuMVtt-EDIII formulated in DOPS indices highest total IgG responses. The figure shows IgGs level of three different groups of mice vaccinated with EDIII alone formulated in phosphate-buffered saline (PBS), with vaccine formulated by coupling EDIII to VLP (CuMVtt-EDIII) or formulating CuMVtt-EDIII plus DOPS adjuvant. The assessment was done with sera collected three weeks after each vaccination. The results were analysed using GraphPad Prism software applied to assess the means of three groups by one-way analysis of variance (ANOVA). The values observed in the negative control group (vaccinated with only PBS) were subtracted from the titres of the other experimental groups. Note: * *p* < 0.01, ** *p* < 0.001, *** *p* < 0.0001.

**Figure 3 vaccines-07-00072-f003:**
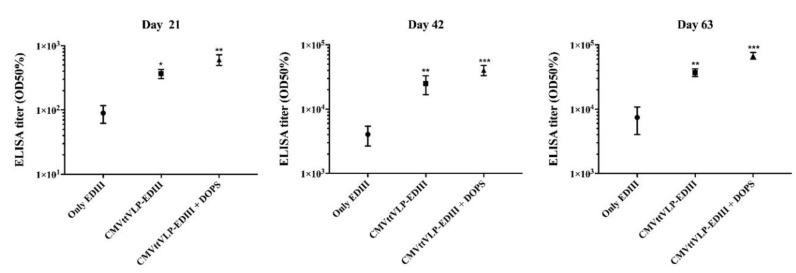
Subclasses of IgG. The figure shows the specific subclasses of IgG induced by each vaccine from samples collected three weeks after the third vaccination (Day 63). The group of mice vaccinated with only EDIII predominantly produced IgG1 while EDIII coupled to VLP (CuMVtt-EDIII) also induced IgG2a and IgG2b. When DOPS adjuvant was ad-mixed, this skewing towards IgG2a and b was even more pronounced. The results were analysed using GraphPad Prism software applied to assess the means of three groups by one-way analysis of variance (ANOVA). The values observed in the negative control group were subtracted from the titres of the other experimental groups. Note: ** *p* < 0.001, *** *p* < 0.0001.

**Figure 4 vaccines-07-00072-f004:**
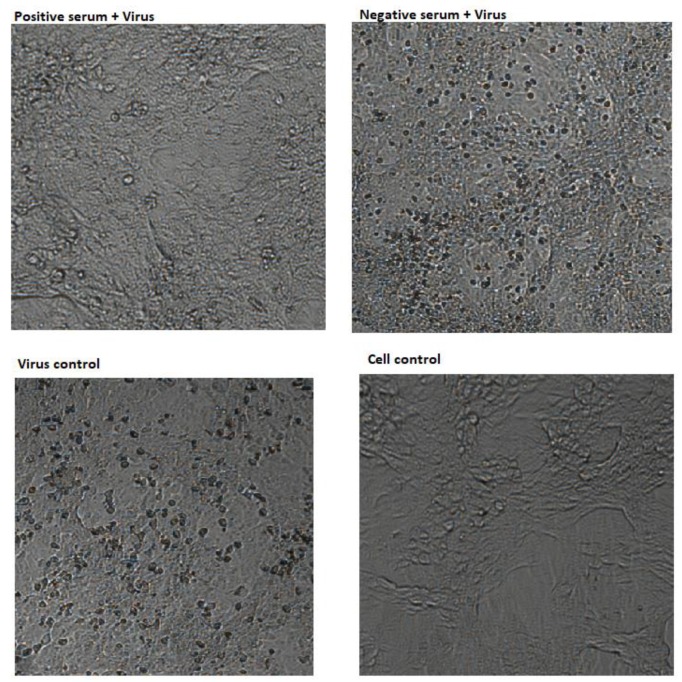
A neutralization test was conducted to evaluate the ability of a ZIKV-specific antibodies to prevent cpe. Cells infected with ZIKV pre-incubated with sera from immunized mice showed no CPA (positive serum + virus). Cytopathic effect (CPE) was seen in serum without ZIKV specific antibodies (Negative serum + Virus), comparable to cultures infected only with ZIKV (Virus control). The panel “Cell control” shows uninfected cells.

**Figure 5 vaccines-07-00072-f005:**
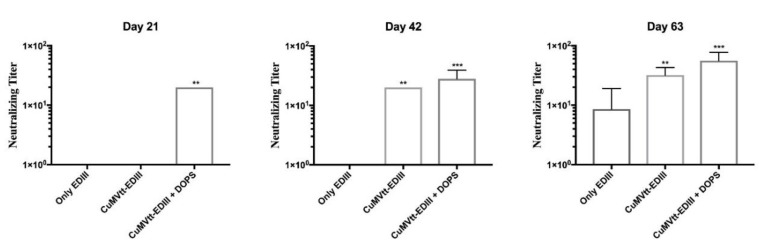
ZIKV neutralising capacity assessment. The figure shows the titer of ZIKV-neutralising antibodies that was evaluated in sera of mice vaccinated with only EDIII protein, CuMVtt-EDIII and CuMVtt-EDIII plus DOPS. The data shown here are from samples collected on days 21, 42 and 63 and two independent experiments were performed. Note: ** *p* < 0.001, *** *p* < 0.0001.

**Figure 6 vaccines-07-00072-f006:**
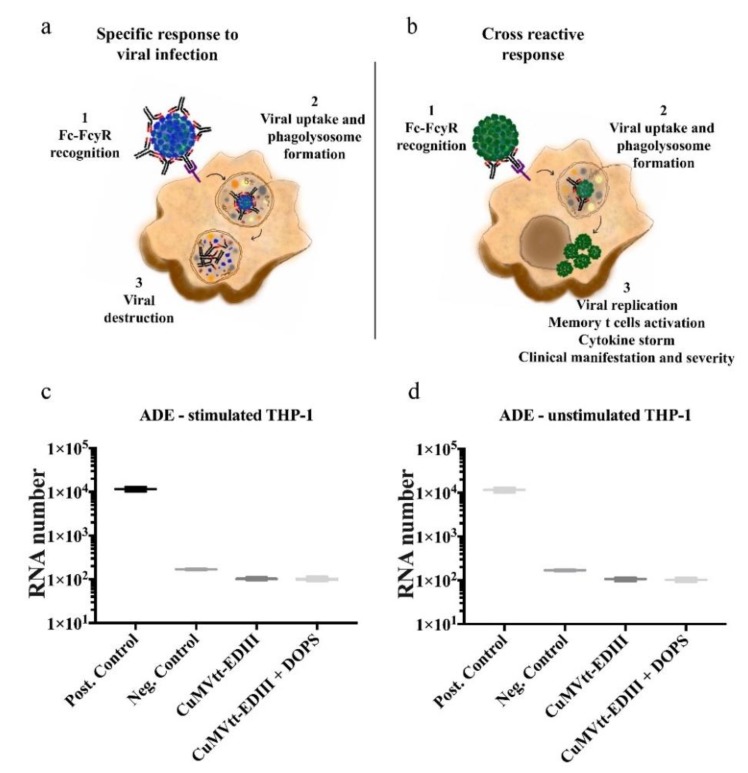
Vaccine-induced antibodies do not cause antibody-dependent enhancement (ADE). A schematic figure representing a specific response against a virus that has been completely opsonised by neutralizing antibody (**a**) and when ZIKV is recognised by non-neutralizing antibodies, the antibody binding will not be to neutralise the *flavivirus*, but instead it will help the viral uptake without destruction via phagolysosomes allowing cellular infection (**b**). The ADE assay was conducted with stimulated (**c**) as well as unstimulated THP-1 cells (**d**). Note: for ADE test, two independent experiments were performed.

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
