# Peer review of "Zika Virus-Derived E-DIII Protein Displayed on Immunologically Optimized VLPs Induces Neutralizing Antibodies without Causing Enhancement of Dengue Virus Infection"

_vaccines, 2019, doi:10.3390/vaccines7030072_

Round 1

Reviewer 1 Report

This manuscript by Cabral-Miranda et al. describes about a development of Zika vaccine candidates which induce neutralizing antibody but not infection-enhancing antibody. Their main strategy is EDIII-based protein vaccine with or without CuMV-VLP. Although EDIII recombinant protein alone induced poor antibody level into mice, combination of EDIII-VLP-adjuvant (CuMVtt-EDIII+DOPS) significantly increased the antibody level. The proportion of IgG subclass was IgG1>IgG2a=IgG2b in EDIII vaccine alone, while CuMVtt-EDIII+DOPS displayed IgG1<IgG2a=IgG2b. The present vaccine system is well organized and can promise as an ideal vaccine candidate against Flaviviruses. However, there are some points which the authors need to clarify. Specific and minor comments are described below.

Specific comments

Is there any reason why you measured each level of IgG subclasses? How does the IgG subclass difference affect antibody activity (neutralizing activity and enhancing activity)? In my knowledge, mouse IgG1 is complement independent antibody, whereas IgG2a and IgG2b are complement dependent ones. Thus, IgG2a and IgG2b may show stronger neutralizing activity in the presence of complement. I am interested in seeing a result of neutralization test in which complement is added (Figure 4). Probably, it is expected that the neutralization titer shown by CuMVtt-EDIII(+DOPS) will be much higher than that by only EDIII.

In Figure 5, they analyzed ADE activity against DENV-2 only. However, it is not sufficient to demonstrate “no enhancement of dengue virus infection”. At least, ADE activity against four serotypes of DENV should be measured.

Minor comments

Last sentence in Methods section 2.6: Although they described that “ADE was calculated as fold-change increase…”, Y-axis in Figures 5c and 5d indicates RNA number.

Figures 5c and 5d: Please specify the positive control.

Figures 3 and 5: What day’s samples were used in these results?

Author Response

Cover letter explaining point-by-point the revisions in the manuscript and responses to the comments

We would like to thank you for the valuable comments and suggestions you have made. All our responses (in black) to your questions (in blue) may be found below. We hope the modified manuscript may now be acceptable for publication in Vaccines.

Comments and Suggestions for Authors

This manuscript by Cabral-Miranda et al. describes about a development of Zika vaccine candidates which induce neutralizing antibody but not infection-enhancing antibody. Their main strategy is EDIII-based protein vaccine with or without CuMV-VLP. Although EDIII recombinant protein alone induced poor antibody level into mice, combination of EDIII-VLP-adjuvant (CuMVtt-EDIII+DOPS) significantly increased the antibody level. The proportion of IgG subclass was IgG1>IgG2a=IgG2b in EDIII vaccine alone, while CuMVtt-EDIII+DOPS displayed IgG1<IgG2a=IgG2b. The present vaccine system is well organized and can promise as an ideal vaccine candidate against Flaviviruses. However, there are some points which the authors need to clarify. Specific and minor comments are described below.

We would like to thank you for your valuablecomments. All of your questions are responded to below in detail.

Specific comments

Is there any reason why you measured each level of IgG subclasses? How does the IgG subclass difference affect antibody activity (neutralizing activity and enhancing activity)? In my knowledge, mouse IgG1 is complement independent antibody, whereas IgG2a and IgG2b are complement dependent ones. Thus, IgG2a and IgG2b may show stronger neutralizing activity in the presence of complement. I am interested in seeing a result of neutralization test in which complement is added (Figure 4). Probably, it is expected that the neutralization titer shown by CuMVtt-EDIII(+DOPS) will be much higher than that by only EDIII.

We thank the reviewer for this comment.

The reason why we wanted to understand the subclass distribution is because that knowledge will help us to deduce the characteristics of the immune response associated with vaccination, such as Th1 vs Th2, functional features etc. Therefore, we agree with the reviewer that IgG subclasses that trigger complement-dependent, opsonization and lysis may result in higher neutralizing titers. Although such a test has been reported for several viruses, including flaviviruses, it has also been shown that complement alone can inactivate WNV and we have shown that DENV-2 and ZIKV are neutralized directly by complement (unpublished). Therefore, we believe that in this case a complement dependent neutralization assay will not have an added value as we will not be able to distinguish between direct and antibody-dependent complement mediated virus-neutralisation. But we agree that more efforts should be deployed to find a source of complement that does not directly inactivate ZIKV and can be used to study this interesting question. We have added all of this information to the discussion (highlighted in yellow). 

In Figure 5, they analyzed ADE activity against DENV-2 only. However, it is not sufficient to demonstrate “no enhancement of dengue virus infection”. At least, ADE activity against four serotypes of DENV should be measured.

We agree with the reviewer that our results only indicate that ZIKV-induced antibodies do not enhance DENV-2/NGC strain. However we cannot exclude that other serotypes could be enhanced. On the other hand, it is important to call attention that DENV-2 has been proposed to be the most virulent DENV serotype, besides that, the most epidemics of dengue haemorrhagic fever have been associated with DENV-2 rather than the other serotypes, it has been the most genetically diverse population, and it is the most frequent cause of dengue epidemic worldwide, which is known to be associated with severe dengue cases. Therefore, the use of DENV-2 appears therefore the best and most significant model for this study. But we agree that other studies investigating enhancement activity of flavivirus cross-reactive antibodies must include a panel of DENV-serotypes and genotypes. This has now been discussed in the results and discussion (highlighted in yellow) to put our results into perspective.

Minor comments

Last sentence in Methods section 2.6: Although they described that “ADE was calculated as fold-change increase…”, Y-axis in Figures 5c and 5d indicates RNA number.

We acknowledge your comment. It is now fixed in the manuscript, Method section 2.6. and highlighted in yellow.

Figures 5c and 5d: Please specify the positive control.

It is now specified in the manuscript, Method section 2.6. and highlighted in yellow.

Figures 3 and 5: What day’s samples were used in these results?

Thank you for your observation. This information are now provided in the manuscript and highlighted in yellow. 

Reviewer 2 Report

In this manuscript, authors introduce a ZIKV EDIII targeted vaccine candidate based a excellent vaccine vector-VLP and vaccine formulation. Following items as concerns:

1. How about the uniformity of the quality of formulation CuMVVLP-EDIII?

2. Suggest to test the neutralizing Abs induced by CumVVLP-EDIII for DENV.

3. According to current reported references(PMID: 29362446, PMID: 28669618, and PMID: 29665376 ) for ZIKV EDIII, the neutralizing Abs titers are variable greatly, is there possible to add challenge results for this study, and try to discuss in discussion part.

4. For the RT-PCR, please describe the method, and indicate the detect limitation. 

Author Response

Cover letter explaining point-by-point the revisions in the manuscript and responses to the comments

We would like to thank you for the valuable comments and suggestions you have made. All our responses (in black) to your questions (in blue) may be found below. We hope the modified manuscript may now be acceptable for publication in Vaccines.

Comments and Suggestions for Authors

In this manuscript, authors introduce a ZIKV EDIII targeted vaccine candidate based an excellent vaccine vector-VLPs and vaccine formulation. Following items as concerns:

1. How about the uniformity of the quality of formulation CuMVVLP-EDIII? 

Thank you for your comment. It is important to note that, the conjugation of antigens to CMVttVLP using the chemical cross-linker SMPH (succinimidyl-6-(b-maleimidopropionamido) hexanoate)  and SATA (N-succinimidyl-S-acetylthioacetate) is a very well tested protocol used by our group and tested and established for many other antigens. In addition, we analyze here for the first time vaccine formulations (CuMVttVLP-E-DIII) as well as CuMVttVLP, by Atomic force microscopy (AFM).

2. Suggest to test the neutralizing Abs induced by CumVVLP-EDIII for DENV.

Thank you for your suggestion. However, the most important experiment to test cross-reactivity between anti-ZIKV and DENV is the possible enhancement of infection. And this experiment has been provided.

3. According to current reported references (PMID: 29362446, PMID: 28669618, and PMID: 29665376 ) for ZIKV EDIII, the neutralizing Abs titers are variable greatly, is there possible to add challenge results for this study, and try to discuss in discussion part.

We acknowledge your suggestions. For better explanation about our results we inserted more information into the manuscript and discussed that (highlighted in yellow). Unfortunately, we are currently not equipped to perform in vivo challenge results. However, we do not think that this would make the results more uniform.

4. For the RT-PCR, please describe the method, and indicate the detect limitation. 

Thank you for your valuable suggestion. A new reference was inserted, as well as a better explanation is described into the M&M. The limit of detection for the assay that used is 200 copies/ml (all information are highlighted in yellow).

Reviewer 3 Report

In this manuscript, Cabral-Miranda et al describe the generation and initial serological characterization of a ZIKV EDIII vaccine candidate. Although the concept of an EDIII-based vaccine to elicit a potently-neutralizing, type-specific antibody response is not new, the novelty in their approach lies in the expression of the EDIII vaccine in the cucumber mosaic virus VLP background. 

I have several major reservations about the data presented and methodology of the study performed. These are listed below for the authors' consideration. 

1) The ELISA titers shown in Figures 2 and 3 should be shown in the log scale. Were any EDIII antibodies observed in the EDIII only animals? If so is this value real or background signal? Were any un-immunized animals used as a control group?

2) Determining neutralization titer by assessing CPE is rather odd, and not the typical way in which neutralization assays are performed for DENV and ZIKV. This is because several DENV and ZIKV strains do not cause visible CPE. If the viral strain used by the authors does cause CPE, a representative panel showing uninfected cells vs. infected cells vs. cells treated with serum should be shown as part of Figure 4. 

Was a positive control used here? What negative control serum was used by the authors?

Futher, the legend under Figure 4 states: "The level of capable neutralising dilution is defined by the redline" -- what does this mean? Please clarify.  

3) Please clarify what positive and negative controls were used in Figure 5, and change the axis labels in part c and d to "RNA copy number". Please provide a citation supporting the claim that the ADE-mediated uptake of virus can cause memory T cell activation and cytokine storm -- my understanding is that the link between ADE and disease severity is based on certain hypotheses and has not necessarily been demonstrated as cause-effect in humans.  In addition, there is no evidence to support the claim that ZIKV antibodies enhance DENV infection. Please comment on this topic in the discussion section. 

4) The authors must discuss the obvious limitations of using a BALB/c mouse model for their immunogenicity studies given that BALB/c mice are not readily infected with ZIKV and show no clinical signs of disease (as shown by other studies).

5) In section 2.6 of the methods, the authors state "To measure ADE, RNA was isolated from the supernatant and PCR was conducted as previously described (42)". Firstly, a PCR cannot directly be performed on RNA. Secondly, the paper referenced here does not explicitly describe the conditions used for the qRT-PCR. Please include a direct reference to the paper which describes the primers and specific conditions required for RNA quantification in the ADE assay. 

5) Overall I believe the manuscript would benefit from major editorial assistance. There are grammar and spelling errors throughout the text, and the very brief discussion section leaves much to be desired. Please also make sure that the citations are formatted properly - from a quick look at the references I found that the same papers by Priyamvada et. al and Stettler et al. have been listed twice as #15 and #26, and #19 and #28 respectively. 

Author Response

Cover letter explaining point-by-point the revisions in the manuscript and responses to the comments

We would like to thank you for the valuable comments and suggestions you have made. All our responses (in black) to your questions (in blue) may be found below. We hope the modified manuscript may now be acceptable for publication in Vaccines.

Comments and Suggestions for Authors

In this manuscript, Cabral-Miranda et al describe the generation and initial serological characterization of a ZIKV EDIII vaccine candidate. Although the concept of an EDIII-based vaccine to elicit a potently-neutralizing, type-specific antibody response is not new, the novelty in their approach lies in the expression of the EDIII vaccine in the cucumber mosaic virus VLP background. I have several major reservations about the data presented and methodology of the study performed. These are listed below for the authors' consideration. 

1) The ELISA titers shown in Figures 2 and 3 should be shown in the log scale. Were any EDIII antibodies observed in the EDIII only animals? If so is this value real or background signal? Were any un-immunized animals used as a control group?

Thank you for your observations. The Figures were organized in a way that would be easier and more precise to interpret and compare the groups. We always included a control group of mice vaccinated with only PBS. And the results obtained with PBS control group are deducted from the values obtained with immunized mice. Therefore, all data shown in this manuscript are values above background. This is now better explained in the manuscript (highlighted in yellow).

2) Determining neutralization titer by assessing CPE is rather odd, and not the typical way in which neutralization assays are performed for DENV and ZIKV. This is because several DENV and ZIKV strains do not cause visible CPE. If the viral strain used by the authors does cause CPE, a representative panel showing uninfected cells vs. infected cells vs. cells treated with serum should be shown as part of Figure 4. 

Thank you for your valuable observation. We agree that not all DENV strains cause CPE or even plaques. The situation is different for ZIKV. The majority of ZIKV strains cause CPE on vero cells. Although, it is very much dependent on the source of vero (with inter-lab variation being noted) and passage number. The strain that we have used produces good CPE on vero cells. We have added representative pictures as requested by the reviewer.

3) Was a positive control used here? What negative control serum was used by the authors?

We have used mice that were infected with rabies (historical control) as negative control. This information is inserted in the manuscript and highlighted in yellow.

4) Futher, the legend under Figure 4 states: "The level of capable neutralising dilution is defined by the redline" -- what does this mean? Please clarify.  

Thank you for your observation. We have improved the figure and adapted for a better understanding based on your notification and from other reviewer as well. 

5) Please clarify what positive and negative controls were used in Figure 5, and change the axis labels in part c and d to "RNA copy number". Please provide a citation supporting the claim that the ADE-mediated uptake of virus can cause memory T cell activation and cytokine storm -- my understanding is that the link between ADE and disease severity is based on certain hypotheses and has not necessarily been demonstrated as cause-effect in humans.  In addition, there is no evidence to support the claim that ZIKV antibodies enhance DENV infection. Please comment on this topic in the discussion section. 

In this assay the pan-flavivirus monoclonal antibody (MAB10216, clone D1-4G2-4-15; Millipore, Germany) was used as a positive control whereas a pool of serum from mice infected with rabies virus was used negative control. These information have now been provided in the materials & methods of the revised manuscript and highlighted in yellow.

There is clear evidence in vitro that ZIKV-specific antibodies can enhance DENV-infection. Indeed, antibodies specific for domains I and II are mostly enhancing infection while antibodies against domain III are mostly neutralizing. For the direct clinical link between ADE and enhanced T cell response, the reviewer is right; it`s mostly a hypothesis. We have adjusted the text accordingly and provided more references about.

6) The authors must discuss the obvious limitations of using a BALB/c mouse model for their immunogenicity studies given that BALB/c mice are not readily infected with ZIKV and show no clinical signs of disease (as shown by other studies).

The reviewer is right in saying that BALB/c mice cannot be infected with ZIKV. However, both BALB/c and C57BL/6 have been used extensively by immunologists to study immunogenicity of vaccines. Although different strains have predominance for a cytokine profile (Th1 vs Th2 skewed), immunocompetent mice, including BALB/c are valid models to investigate immunogenicity. However, we agree that challenge studies must be performed in susceptible mice, of which the AG129 is the most reported one. The goal of our study was to compare immunogenicity of different vaccine candidates and the capacity of Ab response against each vaccine to neutralize the virus as well. Besides that, another important goal was to analyse the possibility of ADE with DENV.

7) In section 2.6 of the methods, the authors state "To measure ADE, RNA was isolated from the supernatant and PCR was conducted as previously described (42)". Firstly, a PCR cannot directly be performed on RNA. Secondly, the paper referenced here does not explicitly describe the conditions used for the qRT-PCR. Please include a direct reference to the paper which describes the primers and specific conditions required for RNA quantification in the ADE assay. 

Thank you for your comments. A better description of the PCR as well as other references were added to the manuscript and highlighted in yellow.

8) Overall I believe the manuscript would benefit from major editorial assistance. There are grammar and spelling errors throughout the text, and the very brief discussion section leaves much to be desired. Please also make sure that the citations are formatted properly - from a quick look at the references I found that the same papers by Priyamvada et. al and Stettler et al. have been listed twice as #15 and #26, and #19 and #28 respectively. 

We acknowledge your comments and observations.  The manuscript was improved in term of grammar, as well as the results and discussions were extended to better understand. Besides that, the references are now properly formatted. 

Reviewer 4 Report

The authors describe a use of a technology they recently developed to couple Zika's EDIII domain on VLPs to generate a humoral response in mice. The sera were then tested for recognition (total and subtype IgG), after neutralization tests they tested for an absence of ADE with dengue in vitro (THP1).

This an interesting work.

Major comments:

In Figure 4 the neutralisation titer is not high. Would this immunisation be sufficient to provide protection against ZIKV infection? What happens in a deficient model (IFN deficient) in terms of protection conferred by immunisation with VLP-EDIII followed by a challenge with ZIKV?

Minor comments:

1/For EDIII preparation and purification 8M urea is used to solubilize the protein. The protein is denatured, renaturation took place during the filtration gel? or before? the buffer used during the filtration gel must be indicated and the method of renaturation/removal of the urea too.

2/Subtitles are sometimes underlined and sometimes not, please be consistent.

3/Ethics statement and approval can be inclued in section 2.3?

4/In fig. 2 and 4 "ELISA titre" should not be "ELISA titer"?

5/In section 3.4 you talk about cross-reactivity in the experiments presented in Fig. 5? You have tested for ADE but not for cross-reactivity to my point.

6/Statical analysis is quite confusing, please indicate what you are comparing for each graph (Fig 2 and 3).

7/There is no statistical analysis for fig. 4 and 5? These experiments were repeated n=? and stats?

8/The legend in Fig. 5 corresponds to a results section (interpretation) and therefore does not correspond to a legend. please rewrite.

9/ Flaviviruses have PS on their surface. Can the increase in titer and neutralization not be due to the presence of anti-PS antibodies (presence of DOPS as an adjuvant)?

Author Response

Cover letter explaining point-by-point the revisions in the manuscript and responses to the comments

We would like to thank you for the valuable comments and suggestions you have made. All our responses (in black) to your questions (in blue) may be found below. We hope the modified manuscript may now be acceptable for publication in Vaccines.

Comments and Suggestions for Authors

The authors describe a use of a technology they recently developed to couple Zika's EDIII domain on VLPs to generate a humoral response in mice. The sera were then tested for recognition (total and subtype IgG), after neutralization tests they tested for an absence of ADE with dengue in vitro (THP1). This is an interesting work. 

We acknowledge you for the comment.

Major comments:

In Figure 4 the neutralisation titer is not high. Would this immunisation be sufficient to provide protection against ZIKV infection? What happens in a deficient model (IFN deficient) in terms of protection conferred by immunisation with VLP-EDIII followed by a challenge with ZIKV?

Comparing with the literature, it seems that the titer of Figure 4 is within the published range, albeit at the lower end. Besides, also in humans infected with ZIKV, relatively low VN titers are found. This is better discussed now in the manuscript and highlighted in yellow.

Minor comments:

1) For EDIII preparation and purification 8M urea is used to solubilize the protein. The protein is denatured, renaturation took place during the filtration gel? or before? the buffer used during the filtration gel must be indicated and the method of renaturation/removal of the urea too.

Thank you for your observation. A better explanation is now described in the section “2.1 ZIKV E-DIII Protein Production” and highlighted in yellow.

2) Subtitles are sometimes underlined and sometimes not, please be consistent.

We acknowledge you for this important observation. Now all the Subtitles are underlined. 

3) Ethics statement and approval can be include in section 2.3?

Actually it is a good idea, to include this section before describe any animal use. Thank you!

4) In fig. 2 and 4 "ELISA titre" should not be "ELISA titer"?

It is ok to right in both ways, however as it was written “titer” in the manuscript, it makes sense to white in the figures as well. It is now fixed. 

5) In section 3.4 you talk about cross-reactivity in the experiments presented in Fig. 5? You have tested for ADE but not for cross-reactivity to my point.

We left only ADE in the sentence and deleted cross-reactivity. You are right, it is more precise and better to understand.

6) Statistical analysis is quite confusing, please indicate what you are comparing for each graph (Fig 2 and 3). 

Thank you for your comment. To better understanding, we described the analysis on the figures’ description and highlighted in yellow.

7) There is no statistical analysis for fig. 4 and 5? These experiments were repeated n=? and stats?

Thank you for your observations. To better comprehension, the statistical analysis was performed.Two independent experiments were performed and all of these informations are provided in the Manuscript and highlighted in yellow.

8) The legend in Fig. 5 corresponds to a results section (interpretation) and therefore does not correspond to a legend. please rewrite.

We, unfortunately, do not understand the reviewers point. To us, the legend seems fine and there is no interpretation in it.

9) Flaviviruses have PS on their surface. Can the increase in titer and neutralization not be due to the presence of anti-PS antibodies (presence of DOPS as an adjuvant)?

PS not chemically linked to a protein are not expected to induce antibody responses, as Th is lacking. Furthermore, the neutralisation titer parallels the ELISA titer. And the ELISA was performed using recombinant EDIII domain, not containing an PS.

Round 2

Reviewer 1 Report

Authors satisfied all reviewer's comments.

Author Response

We appreciate that we satisfied all the reviewers comments and acknowledge all the valuable comments and suggestions you have made to improve this manuscript before be published in the this valuable Journal.

Reviewer 2 Report

no more concerns

Author Response

(The authors gave the same response as above.)

Reviewer 3 Report

While the authors have made several changes based on the reviewers' comments, there are still a few outstanding issues of concern:

1) I am not convinced that a 1.5-2-fold difference in ELISA titers between the +/- DOPS groups is significant from an immunological standpoint -- in fact the dilution series for the ELISA is 1:3, so in all reality, some of these "significant" differences fall between one dilution of serum. This is why I suggested plotting titers in the log scale, which is the standard way of plotting ELISA and neutralization titers. The authors may have declined to replot the data because the difference is not as apparent in the log scale, and will not fit their narrative that +DOPS is the better candidate. 

2) Figure 6 legend: please remove text describing the process of ADE from the figure legend. You may include this description in your results section as a way to introduce parts C and D of the figure. Please simply state what the figure shows, removing commentary such as "highly significant" from the legend. 

Author Response

Cover letter explaining point-by-point the revisions in the manuscript and responses to the comments

We would like to thank you for the valuable comments and suggestions you have made. All our responses (in black) to your questions (in blue) may be found below. We hope the modified manuscript may now be acceptable for publication in Vaccines.

Reviewer:

1) I am not convinced that a 1.5-2-fold difference in ELISA titers between the +/- DOPS groups is significant from an immunological standpoint -- in fact the dilution series for the ELISA is 1:3, so in all reality, some of these "significant" differences fall between one dilution of serum. This is why I suggested plotting titers in the log scale, which is the standard way of plotting ELISA and neutralization titers. The authors may have declined to replot the data because the difference is not as apparent in the log scale, and will not fit their narrative that +DOPS is the better candidate. 

We acknowledge your comment and suggestion. 

As required, all data/figures are plotted in log scale.   

2) Figure 6 legend: please remove text describing the process of ADE from the figure legend. You may include this description in your results section as a way to introduce parts C and D of the figure. Please simply state what the figure shows, removing commentary such as "highly significant" from the legend.

As required by the reviewer, the Fig. 6 legend was improved and some important information were inserted to the results section. 

Reviewer 4 Report

This work can be accepted for publication in vaccines because all my comments from the first round are discussed.

Author Response

(The authors gave the same response as above.)
